



# The Relationship between Tropical Cyclone Activity, Nutrient Loading, and Algal Blooms over the Great Barrier Reef

Chelsea L. Parker[1,2], Amanda H. Lynch[1,2], Stephanie A. Spera[1,2], and Keith R. Spangler[1,2]

[1]Department of Earth, Environmental, and Planetary Sciences, Brown University, Providence RI 02912, USA
    [2] Institute at Brown for Environment and Society, Brown University, Providence RI 02912, USA

*Correspondence to*: Chelsea L. Parker (Chelsea_Parker@brown.edu)

**Abstract.** The Great Barrier Reef, the world's largest coral reef ecosystem, is subject to many environmental stressors. This study utilizes remotely sensed Moderate Resolution Imaging Spectroradiometer (MODIS) chlorophyll *a* concentration data

to explore statistically significant relationships between local-scale tropical cyclone disturbance and relative water quality between 2004-2014. The study reveals that tropical cyclone activity reduces water quality at 8- and 16-day time lags. Relationships suggest that at early stages (during and just after cyclone activity) algal response is induced primarily through wind-driven sediment re-suspension. However, wind speed in isolation only increases minimum levels of chlorophyll *a*, rather than mean or extreme upper values. At greater time lags (16-day), it is suggested that nutrient runoff from rainfall (and

perhaps storm surge) increase phytoplankton activity, leading to detrimental ecological effects. The analyses systematically demonstrate the dominance of tropical cyclone size on mean and extreme values of chlorophyll *a* during and after tropical cyclone activity (at 0-, 8-, and 16-day time lags). Both the total area affected and the area from which nutrients can be extracted have more impact on chlorophyll *a* concentrations than either the duration or intensity of the cyclone. Findings indicate that efforts to reduce nutrient and sediment leaching into the reef lagoon from the Queensland coastal lands need to

be continued and improved. This will be particularly important in the context of climate change, since tropical cyclone frequency, dynamics and characteristics are likely to change.

## 1 Introduction

Spanning 350,000 km$^2$ in area and extending more than 2,300 km along Australia's northeastern coast, the Great Barrier Reef Marine Park encompasses the world's largest coral reef ecosystem (Great Barrier Reef Marine Park Authority 2011;

Stoeckl et al. 2011). The Great Barrier Reef Marine Park is home to over 3000 coral reefs and 1600 fish species, and was added to the World Heritage List in 1981. The park annually generates approximately AUS\$5.7 billion of value added and 69,000 full-time-equivalent jobs for Australia's economy (Deloitte Access Economics 2013).

Despite protection efforts, the Great Barrier Reef Marine Park is still subject to many local- and global-scale stressors associated with a myriad of negative impacts and long-term consequences. Global, long-term pressures include ocean

warming and acidification, resulting in coral bleaching, dissolution, and loss of sensitive reef builders, reduced species richness and abundance, and loss of ecosystem function. This study focuses on the local-scale disturbance of tropical





cyclones and the resultant water quality degradation in the Great Barrier Reef. Such disturbances decrease reef system resilience in the face of larger-scale and longer-term stressors.

Tropical cyclones can reduce water quality over the reef environment by increasing concentrations of suspended sediments, colored dissolved organic matter, nutrients, pesticides, and herbicides in the water column through several possible
mechanisms: (1) wind-driven re-suspension of sediments (in water columns < 50m in depth), (2) nutrient and sediment loading driven by rainfall-induced runoff and river discharge. An overview of the pathways for the two primary mechanisms is demonstrated in Figure 1.

Initially, the chlorophyll *a* concentration in surface waters may respond to the reduced water quality following tropical cyclone activity through the wind-driven re-suspension mechanism. As the tropical cyclone moves over the reef lagoon, the
intense wind and wave energy can re-suspend benthic sediments (Marra et al. 1990; Larcombe and Carter 2004; Wolanksi et al. 2005) with an associated pulse of nutrients from lagoon and reef pore water (Furnas 1989). The re-suspension and resulting nutrient pulse occurs during tropical cyclone passage, and the surface phytoplankton and algal activity respond to the nutrients now available in the surface waters, causing an increase in chlorophyll *a* detection within few days (Furnas 1989). The response in surface chlorophyll *a* levels should be detectable by remote sensing instruments within 10 days of the
wind activity in the lagoon area (Subrahmanyam et al. 2002; Lin et al. 2003; Zheng and Tang 2007; Siswanto et al. 2008). Although remotely sensed chlorophyll *a* concentrations in turbid coastal waters may be confounded by an concurrent influx of total suspended sediments (e.g. Gitelson et al. 2008; Gordon and Morel 2012), it can be inferred that a positive bias in chlorophyll *a* measurements during times with heavy sediment loading would still indicate reduced water quality over the reefs and be collinear with the promotion of chlorophyll *a*.

As the tropical cyclone makes landfall, the coastal land is inundated with convective rain and storm surge. Heavy rain can be sustained for significant distances inland, even as the cyclone weakens. This heavy rainfall results in large river discharges from the Great Barrier Reef catchment into the lagoon, resulting in high concentrations of nutrients and sediment loads (Stieglitz 2005; Brodie 2012b). These nutrients and sediments arise primarily from fertilizer leaching and enhanced erosion (Brodie 2012b). Recent trends in land use can exacerbate these issues. Approximately 40% of the Great Barrier Reef
catchment area has been cleared of natural vegetation (Furnas 2003), and the concentrations of inputs have increased as the density and extent of new land-cover types and activities increase (Hutchings et al. 2005; Kroon et al. 2012). Thus, the Great Barrier Reef lagoon is subject to loading events (Davies and Eyre 2005; Mitchell et al. 2005; Hoegh-Guldberg et al. 2011), with inputs being derived from industrial activity, urban areas, sewage treatment, and agriculture on the coastal lands. Following the tropical cyclone event, nutrient and sediment loading degrades water quality and promotes an increase in
phytoplankton and algal activity, increasing the detected surface chlorophyll *a* concentrations.

The time taken for rivers to discharge to the lagoon remains an open question in this area, owing to the variability of river flow as a function of storm characteristics, basin and catchment geometry and bathymetry, land use, and antecedent conditions. Gong et al. (2007) found that tropical cyclone storm surge retreat to the lagoon may take roughly 3-4 days after a storm event in North Carolina, United States. Alexander et al. (2001) suggested that rainfall runoff from catchment to lagoon





can take roughly one week in parts of Queensland, Australia. The travel time from river mouth to reef lagoon alone requires 1-2 days (Brodie et al. 2010). This sparse evidence leads to expectations that the chlorophyll *a* response to the rainfall discharge would occur at least several days after landfall.

The reduction of water quality through the input of nutrients and sediments through these mechanisms has a negative effect on reef organisms. Sediments and herbicides can inhibit photosynthesis in the benthic coral organisms, causing resource limitation to the phototropic organisms (Devlin and Schaffelke 2009). Nutrients (e.g., particulate nitrogen and phosphorus) can decrease calcification rates (Devlin and Schaffelke 2009), leading to skeletal defects and reproductive impairment (Brodie et al. 2010). Furthermore, an increase in surface photosynthesis through increasing phytoplankton and algal activity (Furnas et al. 2005) can outcompete typical benthic reef producers and builders below (Fraser and Currie 1996; Brodie et al. 2010). The increase in organic matter provided by the phytoplankton is, in turn, cycled through the marine food web and the by-products are deposited on the reef benthic communities, resulting in a number of harmful effects including coral bleaching and mortality through coral smothering (Anthony and Fabricius 2000; Fabricius and Wolanski 2000). Over time, re-occurrence of these long-lasting episodes of degraded water quality can reduce coral species richness (Fabricius and De'ath 2004; Fabricius 2005) and lead to a shift from a coral-dominated to a macroalgal-dominated ecosystem (Fabricius and De'ath 2004; Devlin and Schaffelke 2009; Brodie et al. 2010). This phase shift can cause further mortality of corals and reef organisms (McClanahan et al. 2005; Smith et al. 2006) and would no longer support high biodiversity or provide the same ecosystem services; furthermore, resilience to future disturbances and stressors can be reduced following this phase shift (Lapointe 1997; McCook 1999).

## 1.1 Remote Sensing of Chlorophyll *a*

Chlorophyll *a* is the pigment used by phytoplankton and algae to photosynthesize. Because phytoplankton abundance responds quickly to changes in nutrient availability (typically within 24 hours of nutrient increase; Örnólfsdóttir et al. 2003), chlorophyll *a* concentrations are frequently used as a proxy for phytoplankton biomass and nutrient status in surface waters (Dekker et al. 2006; Furnas et al. 2005; Brodie et al. 2007). Given that chlorophyll *a* is correlated with suspended particulate nitrogen, phosphorus, and other solids in the water column (Fabricius and De'ath 2004; Udy et al. 2005), the Great Barrier Reef Marine Park Authority frequently uses chlorophyll *a* concentrations to monitor and assess the water quality and health of the reef (Brodie and Furnas 1994; Brodie et al. 1997, 2007; Furnas and Brodie 1996; Steven et al. 1998). To explore whether there is a relationship between tropical cyclone activity and water quality over the Great Barrier Reef, chlorophyll *a* concentration data collected over the entire reef expanse and several tropical cyclone seasons are required. Because *in situ* measurements on this spatiotemporal scale are unavailable, we use remotely sensed chlorophyll *a* concentration products to determine and analyze the water quality over ten years across the whole Great Barrier Reef. This approach has been successfully employed in other long-term water quality case studies in a variety of coastal locations, such as in the Chesapeake Bay (Harding and Perry 1997), the Black Sea (Yunev et al. 2002), and the Basque coastal area (Novoa et al. 2012).



Suspended sediments and dissolved organic matter can lead to inaccuracies in remotely sensed chlorophyll *a* estimates in coastal waters (i.e., "Case 2" waters, in contrast to open ocean areas known as "Case 1" waters) when applying global, optical algorithms (e.g., Gordon and Morel 2012; Wang and Shi 2007; Gitelson et al. 2009; and Odermatt et al. 2012).

However, it has been demonstrated that remotely sensed data from the Moderate Resolution Imaging Spectroradiometer (MODIS) are applicable for analysis of coastal waters (Zibordi et al. 2006), and the chlorophyll *a* product can be used to accurately estimate suspended algal activity in coastal areas (Hu 2009). Previous studies have successfully used MODIS chlorophyll *a* data products to analyze chlorophyll changes in other nearshore environments following case study tropical cyclone events (e.g., near Hainan Island; Zheng and Tang 2007; and in the northwest Pacific; Kawai and Wada 2011).

Furthermore, Udy et al. (2005) demonstrated that MODIS chlorophyll *a* data follow a similar trend to *in situ* Great Barrier Reef chlorophyll *a* measurements. That said, MODIS-derived chlorophyll *a* measurements can be problematic in coastal waters during times of high suspended sediment (Gitelson et al. 2008), as would be expected in times of cyclone activity (Larcombe and Carter 2004; Wolanksi et al. 2005). While augmented suspended sediment concentrations may lead to an overestimation of chlorophyll *a* values, the presence of suspended sediments in the water column reduces clarity and has

negative consequences for reef organisms (Russ and McCook 1999; Birrell et al. 2005, Devlin and Schaffelke 2009). The sediments also lead to favorable conditions for phytoplankton growth, given the associated increased nutrient load (Zheng and Tang 2007). In that context, while not unambiguous, positive changes in measured chlorophyll *a* concentrations uniquely represent degraded water quality across the Great Barrier Reef. To account for the potential overestimation of the effects of cyclone activity on chlorophyll *a*, and to ensure our results our robust, this study also considers the relative

difference in chlorophyll *a* concentrations at time lags after tropical cyclone activity, at which time total suspended sediment concentrations in the surface waters may have declined (Brodie et al. 2010).

The median chlorophyll *a* concentration is 0.56 mg m$^{-3}$ globally (Boyce et al. 2010) and ~ 0.46 mg m$^{-3}$ for the Great Barrier Reef (De'ath et al. 2010). However, there are both temporal and spatial variations in these concentrations. Seasonally,

summer values in the Great Barrier Reef average around 0.6 mg m$^{-3}$, while winter values are roughly 0.3 mg m$^{-3}$ (De'ath 2007). Geographically, concentrations in the south and central Great Barrier Reef are approximately 0.54 mg m$^{-3}$, but only about 0.23 mg m$^{-3}$ in the Far North (Brodie et al. 2007). There is additionally a cross-shelf gradient from 1 mg m$^{-3}$ inshore to 0.25 mg m$^{-3}$ offshore (Furnas and Brodie 1996). Although Great Barrier Reef chlorophyll *a* concentrations are considered to be low, relative to the 1-13 mg m$^{-3}$ concentrations in other coastal regions (such as the Gulf of Mexico (Miller and McKee

2004)), water quality degradation from land-based runoff is a major threat to the sensitive environments of the Great Barrier Reef (Haynes and Michalek-Wagner 2000; Haynes et al. 2007; Great Barrier Reef Marine Park Authority 2014).

Previous studies have assessed the spatial and temporal patterns of flood plumes in the Great Barrier Reef lagoons (Devlin 2001; Devlin et al. 2013) and the effect of terrestrial nutrient runoff on the coral organisms specifically (Devlin and Brodie 2005; Wooldridge et al. 2006). In this study, we explore whether a relationship between tropical cyclone activity and relative



water quality can be detected over the entire Great Barrier Reef when accounting for all tropical cyclone and tropical storm activity between 2004-2014. We use statistical tests to address whether significant relationships can be diagnosed at these large temporal and spatial scales between chlorophyll $a$ concentrations across the Great Barrier Reef and tropical cyclone activity and the specific characteristics of wind, rainfall, size, and translation speed. We additionally analyze the water

quality at different time lags to the tropical cyclone activity to investigate the relative dominance of the tropical cyclone mechanisms for reducing water quality.

## 2 Methods

### 2.1 Remotely Sensed Data

MODIS is a multiband imaging spectrometer that was launched by NASA in 2002 on board the Aqua satellite platform. The

Aqua satellite is in a sun-synchronous, near-polar orbit at a 705-km altitude, which allows the MODIS instrument to image the entire Earth every one to two days. MODIS provides quantitative data on global ocean bio-optical properties to examine oceanic factors that affect global change and to assess the role of oceans in the global carbon cycle, as well as other biogeochemical cycles (Ocean Biology Processing Group 2003).

We used the 0.041˚ (~4.6 km at the equator), 8-day Level 3 Aqua MODIS Chlorophyll $a$ concentration product processed by

the NASA Ocean Color Team (Ocean Biology Processing Group 2003). MODIS is an optical sensor and, as such, its ability to collect data can be inhibited by sun glint, inter-orbit gaps, ice, low ambient light, and clouds. Cloud interference presents a considerable impediment when using this product to analyze the relationship between tropical cyclone activity and surface chlorophyll $a$ concentrations, as tropical cyclone activity intrinsically brings cloud cover. By averaging the cloud-free acquisitions at each pixel over an 8-day period, the 8-day product is more robust to these sources of error.

Chlorophyll $a$ concentration is calculated using radiative transfer equations, and the product is derived from the OCI Algorithm (Ocean Biology Processing Group 2003). The algorithm is based on methods developed for the Coastal Zone Color Scanner program (Gordon and Clark 1980) and has been refined and adapted for MODIS bands (Carder et al. 2003). This adaptation has also improved the accuracy of the chlorophyll $a$ estimates in coastal (i.e., Case 2) waters (Carder et al. 2003).

All data composited between 1 January and 15 May (tropical cyclone season for north-eastern Australia) for the years from 2004 to 2014 were downloaded for analysis. We analyzed each global 8-day composite over the Great Barrier Reef region delimited using Geographic Information Systems (GIS) and a shapefile from the Great Barrier Reef Marine Park Authority. The area extends from 10.7°S to 24.6°S and covers an area of 347,192 km$^2$ (Figure 2).

### 2.2 Meteorological Data

We used meteorological data for the independent variables in our statistical analyses, including tropical cyclone presence or absence. Tropical cyclone presence or absence was determined for every 8-day period using the coordinate locations of



storms recorded in the Operational Data for each tropical cyclone season from 2004 to 2014 (Australian Bureau of Meteorology 2014a). For the predictor variables of activity, we included all events from a tropical low to Category 5 storm. The total number of events in the study period is 43, with an annual maximum of seven events and an annual average of four events (Figure 3). Some events bridge more than one 8-day time period, and so there are ~57 time slices that include tropical

cyclone activity (depending on the lag time analyzed). However, a number of these events overlap in time, and so are unlikely to produce a unique chlorophyll *a* response.

We also tested the tropical cyclone characteristics from the Cyclone Damage Potential (CDP) dataset (Holland et al. 2016; see also: Done et al. 2015) for each tropical cyclone that occurred during our study period. The CDP dataset includes 6-hourly maximum wind speeds, translation speed of the system, and radius of 64-knot wind data (i.e., storm size) for tropical

cyclone events of Category 1 (on the Saffir-Simpson Scale) or greater (~15 distinct tropical cyclone events and ~22 time periods with tropical cyclone activity > category 1, depending on the time lag being analyzed). We used this dataset to identify tropical cyclone events affecting the Great Barrier Reef area between March 2004 and early April 2014 (the dates available for this dataset), and then calculated the maxima and means of the maximum wind speeds, translation speeds, and tropical cyclone sizes for each 8-day period to be used as the tropical cyclone characteristics predictor variables of

chlorophyll *a* concentrations.

Daily coastal weather data during our study period were retrieved from the Australian Bureau of Meteorology weather stations at Lockhart River Airport, Cairns Aero, Townsville Aero, Mackay M.O., Gladstone Airport, and Gladstone Radar (Australian Bureau of Meteorology 2014b) (locations depicted with stars on Figure 2). We calculated the maximum and average daily rainfall total, wind speed, and wind gust over each 8-day period when a tropical cyclone was present in the

area. We used the coastal meteorological data to infer the relative importance of the mechanisms for delivering nutrients to the surface lagoon waters during a tropical cyclone event by examining the relative significance and correlation of rainfall and wind activity to chlorophyll *a*.

### 2.3 Statistical Techniques

Using only the pixels that had data in the 8-day composite, we calculated the minimum (MIN), mean, 90th percentile (90

PCT), 95th percentile (95 PCT), and maximum (MAX) of chlorophyll *a* in each 8-day time step over the whole of the Great Barrier Reef. We then calculated the proportion of pixels that exceeded a level of 1 mg m$^{-3}$ (% > 1mg m$^{-3}$) over the study region for each 8-day period. This value is roughly double the threshold suggested by De'ath and Fabricius (2010) for maintaining high coral richness and low macroalgal cover. Pixels in excess of 1 mg m$^{-3}$ indicate a strong potential for damaging effects on the reef environment. These metrics of chlorophyll *a* concentrations represent the dependent variables

of analysis, as we sought to determine whether, and to what extent, different tropical cyclone characteristics influence these concentration metrics. Since tropical cyclones are extreme events and hence rare by definition, we found it important to include a range of descriptive statistics that reflect the spread of the data (e.g., minimum, mean, upper percentiles, and maximum), so that outliers are properly accounted for.





We then assessed separate bivariate and (where applicable) multivariate relationships between each of these descriptive statistics of chlorophyll *a* and each of the meteorological independent variables using the following statistical methods (Table 1): (1) Student's two-sample T-tests to identify significant differences in chlorophyll *a* concentrations between pixels

with and without tropical cyclone activity in the Great Barrier Reef; (2) multivariate one-way ANOVA to estimate the extent to which tropical cyclone characteristics can explain the variance in chlorophyll *a* concentrations, and which characteristics are significant; and (3) single-factor and piece-wise linear regressions to explore the relationships between chlorophyll *a* concentrations and tropical cyclone size, maximum wind speed, translation speed, coastal rainfall, coastal wind, and coastal gust speeds.

We conducted the statistical analyses for several 8-day time lag increments, based on the available 8-day MODIS composites. For this analysis, the time-lag increments (relative to meteorological data) are referred to as follows: the "no-lag" increment refers to the MODIS composite collected concurrently with the meteorological data over an 8-day period; the "8-day lag" refers to the MODIS composite analyzed against meteorological data from the preceding 8 day period. Other lag increments (16-day, 24-day, 32-day) follow a similar pattern. Statistical analyses were first performed for no-lag data, then

repeated for 8-day, 16-day, 24-day, and 32-day lag periods in order to infer possible causal mechanisms between tropical cyclone and phytoplankton activity in the coastal lagoon. We also estimated the response time of the phytoplankton to the tropical cyclone activity, the time of maximum phytoplankton growth and lowest water quality, and the temporal extent of the effect of tropical cyclone activity on the chlorophyll *a* concentrations.

Critical to the interpretation of the mechanisms governing water quality change is the recognition that if the peak remotely

sensed chlorophyll *a* response was found during the time of tropical cyclone activity or shortly thereafter (roughly 1 week or less, given results from Subrahmanyam et al. 2002; Lin et al. 2003; Siswanto et al. 2008), then we could infer that the phytoplankton activity was primarily excited by the wind-driven re-suspension mechanism. If, however, the peak response was not detected within this short lag (roughly ~1 week), then it is more likely that the rainfall inundation and river discharge was the dominant mechanism for providing nutrients to augment phytoplankton activity (in the context of results from

Alexander et al. 2001 see Figure 1).

### 2.4 Hypothesized Relationships

As the tropical cyclone passes over the reef and coastal lagoon (0-8 day time lag), it can be hypothesized that wind speeds have a positive correlation with chlorophyll *a* (especially the mean and extreme values) due to re-suspension of benthic sediments and nutrient pulses from pore waters (Furnas 1989). During this period in which wind-driven re-suspension

dominates, a negative relationship would be detected between translation speed and chlorophyll *a*. If a tropical cyclone moves more slowly across the Great Barrier Reef lagoon, then the region experiences tropical cyclone wind conditions for a longer duration, allowing for increased re-suspension of sediments. However, the inverse may also be possible, where chlorophyll *a* is positively correlated with tropical cyclone translation speed because an increased translation speed increases



the ground-relative wind speed, in turn promoting sediment and associated nutrient re-suspension and a concomitant phytoplankton response. A positive correlation of tropical cyclone size with chlorophyll *a* can be hypothesized: as tropical cyclone size increases, the total area that sediments can be re-suspended from the wind action increases. At this time scale, strong correlations between rainfall and chlorophyll *a* measures may be unlikely given the time required for flow through the

catchment to the lagoon.

After the tropical cyclone makes landfall, elevated phytoplankton mass due to the wind-suspended nutrient loading may persist (Lin et al. 2003). However, we hypothesize that a higher magnitude response will be more closely correlated with the cyclone characteristics that control the ensuing rainfall inundation and river discharge mechanism. At the time scale of 8-day and 16-day lags, a significant positive correlation between rainfall and chlorophyll *a* would persist, whereby rainfall

transports nutrients and sediments to the Great Barrier Reef lagoon through increased runoff and river discharge. These increases would result in more chlorophyll *a* detected in the lagoon. The influence of rainfall on chlorophyll *a* may be stronger than wind speed at this timescale, since wind-driven re-suspension only occurs during tropical cyclone passage over the lagoon. We also anticipate a negative influence of translation speed on chlorophyll *a*. Maximum potential rainfall in a location affected by tropical cyclone activity is typically inversely proportional to the system's translation speed (Elsner and

Kara 1999) and faster forward motion reduces the risk of devastating floods (Elsner and Kara 1999). A positive relationship between tropical cyclone size and chlorophyll *a* is expected, particularly at the mean and extreme upper values. As tropical cyclone size increases, the total catchment area exposed to heavy rainfall typically increases and the storm surge and coastal flood inundation volumes also increase (Irish et al. 2008). Subsequently, the positive correlation between these physical effects and nutrient and sediment loading to the Great Barrier Reef lagoon also intensify, ultimately resulting in reduced

water quality.

## 3 Results and Discussion

The spatial distribution of chlorophyll *a* concentrations over the summer season in the 10-year study period is given in Figure 4 to contextualize the analysis that follows. The average chlorophyll *a* in Figure 4a shows a cross-shelf gradient similar to that reported in Furnas and Brodie (1996), with at least 1 mg m$^{-3}$ average concentrations in the coastal lagoons and

around some of the southern reef atolls and decreasing away from the coastline. Extreme water quality reductions are evident (yellow to brown colors, Figure 4b) in lagoons associated with larger urban centers such as the Cairns to Townsville lagoon, and the Gladstone lagoon, all of which have maximum values far exceeding the 1 mg m$^{-3}$ threshold. The Far North region is often considered the most pristine area of the Great Barrier Reef for water quality (De'ath and Fabricius 2010). However, our results demonstrate that the lagoon just south of Lockhart also has elevated chlorophyll *a* and the water quality is a cause

for concern for reef health in this location. Although this lagoon does not border a dense urban area, the Far North region is known for cattle grazing, sugarcane agriculture, and tropical fruit horticulture (Agricultural Land Audit 2013); hence, nutrients and pesticides leached from these areas during times of tropical cyclone activity and heavy rain exacerbate the reduction in water quality.



### 3.1 Tropical Cyclone Activity

Our analysis demonstrates a significant influence of tropical cyclone activity on water quality in the Great Barrier Reef. In particular, we found a statistically significant elevation in chlorophyll *a* concentrations at 8-day and 16-day lags after tropical cyclone activity. At no lag and lags longer than 16 days, no statistically significant elevations were observed. The peak

differences occur at an 8-day time lag (Table 2). Specifically, mean values of chlorophyll *a* were systematically greater when a tropical cyclone was present in the preceding 8-day period, with median values increasing by about 9% from ~0.47 mg m$^{-3}$ with no tropical cyclone and ~0.51 mg m$^{-3}$ with tropical cyclone and a large increase in the spread of the data during tropical cyclone presence (Figure 5a). More striking, though, is the pattern for the extreme upper values of chlorophyll *a* measures at the 16-day time lag where maximum values are significantly elevated (Figure 5b). The Student's T-test confirms that a

significant difference of 8.06 mg m$^{-3}$ exists between these means (Table 2). Augmented phytoplankton response is therefore still detected over two weeks after the passage of the tropical cyclone. This peak in chlorophyll *a* concentrations at both 8- and 16-day time lags following tropical cyclone activity concurs with the findings of Alexander et al. (2001), suggesting that the process of nutrient and sediment delivery from catchment discharge is the dominant mechanism for reduced water quality. If the effect had been significant from no time lag to an 8-day lag, then this would have been indicative of a more

immediate wind-driven re-suspension mechanism dominance (see again Lin et al. 2003). Preliminary results, then, suggest an influence of tropical cyclone activity on chlorophyll *a* arising from the secondary consequences of terrestrial flooding.

### 3.2 Wind-Driven Re-suspension (No Lag and 8-Day Lag)

To explore the discrete mechanisms by which tropical cyclone activity affects Great Barrier Reef water quality, we looked more closely at the relationship between specific tropical cyclone characteristics and chlorophyll *a* concentrations. During

tropical cyclone activity and just after (no-lag and 8-day lag), tropical cyclone characteristics collectively are found to be significantly related to the variance of chlorophyll *a* concentrations in the Great Barrier Reef, with tropical cyclone size as the dominant characteristic influencing water quality. During the tropical cyclone event (no lag), storm characteristics explain 55-65% of the chlorophyll *a* measures, particularly the mean, upper percentiles, and proportion of pixels that exhibit exceedance of water quality threshold (% > 1 mg m$^{-3}$), as indicated by the multivariate analysis of variance (see Table 3 for

ANOVA results). While tropical cyclone wind speed significantly accounts for up to 14% of the predictive power of this model for the mean and 90th percentile of chlorophyll *a*, neither translation speed nor rain proportions are significant in this timeframe (Table 3). Therefore, at this timescale, the re-suspension mechanism accounts for a larger portion of the detected change in chlorophyll *a* than rainfall runoff and river discharge. The tropical cyclone size dominates the model and significantly accounts for up to 60% of the predictive power (Table 3). This suggests that the total area affected by tropical

cyclone conditions is the most influential characteristic for altering the chlorophyll *a* concentration at this timescale.

Contrary to the hypothesis, there are few significant direct relationships between tropical cyclone wind speed and chlorophyll *a* dependent variables during tropical cyclone activity (no lag and 8-day lag), except for minimum chlorophyll *a*





values over the Great Barrier Reef (Table 5a, b). The only significant positive correlations are between tropical cyclone wind speed and minimum chlorophyll *a* at an 8-day time lag ($r^2$=0.262, p=0.018; see Table 5b), and between the minimum chlorophyll *a* and average weather station gust speed recorded at no time lag ($r^2 = 0.135$, p=0.007; Figure 6a). This suggests that increased tropical cyclone and coastal recorded wind speeds only affect minimum and background values of chlorophyll

*a*. While *greater* wind speeds would result in the re-suspension of benthic substrates and pore water nutrients (Furnas 1989), the Great Barrier Reef lagoon is known to be a naturally nutrient-limited ecosystem (Furnas 2003; Furnas et al. 2005; Dekker et al. 2006). Hence, the concentration of re-suspended nutrients and the corresponding phytoplankton response would both be small. These findings support earlier results that Great Barrier Reef water quality is unlikely to be influenced by static or re-suspended concentrations of sediments, nutrients, or pollutants already in reef environments (Furnas 2003).

During the 0- and 8-day lag increments relative to tropical cyclone activity, we do not find any statistically significant relationship between tropical cyclone translation speed and chlorophyll *a* (Table 5a, b and Figure 7a, b). Neither the duration of exposure to tropical cyclone winds nor increased ground-relative wind speed exhibit a strong influence at this timescale. The weak response of the chlorophyll *a* to the wind speed characteristic may explain the lack of response to translation speed, since the rate at which the tropical cyclone travels primarily influences the experienced wind speed intensity and

duration.

During the passage of the tropical cyclone (no lag), tropical cyclone size has a strong positive influence on chlorophyll *a* concentrations, as reflected in measures of mean, top decile, and proportion of area exceeding the 1 mg m$^{-3}$ threshold for water quality (Table 5a; Figure 8a). Note the appearance of clustering on either side of a potential breakpoint at ~35 km radius in Figure 8a. This cluster pattern is also present in the relationship between cyclone size and other chlorophyll *a*

measures at no lag. We tested piece-wise regressions considering the individual clusters, which do not yield significant relationships between tropical cyclone size and chlorophyll *a*. However, we can infer that the positive correlation in the overall data may be driven by the difference between the clusters, if they are representing a physically realistic breakpoint. At no time lag, the relationship between tropical cyclone size and chlorophyll *a* may not be directly continuous, but could be categorical, whereby larger tropical cyclones (>35 km) lead to increased measures of chlorophyll *a* than smaller tropical

cyclones (<35 km). More data is required to differentiate between the continuous or categorical positive correlation. Regardless, increasing tropical cyclone size increases the total area that nutrients can be extracted from by tropical cyclone conditions (such as wind speed). The data therefore suggest that total area affected by tropical cyclone conditions is more important than the magnitude of the duration and intensity of the exposure of the Great Barrier Reef to tropical cyclone wind conditions.

The positive correlations between chlorophyll *a* and tropical cyclone size (and the dominance of this characteristic) are maintained at the 8-day time lag (see Table 5b). However, the positive correlations are weaker. For example, the correlations with the chlorophyll *a* mean and the % of pixels > 1 mg m$^{-3}$ are reduced to $r^2$=0.185 (Table 5b) and $r^2$=0.2705 (Table 5b, Figure 8b) respectively. This is also reflected in the multivariate ANOVA where, collectively, the tropical cyclone characteristics no longer significantly explain the variance in the chlorophyll *a* at the 8-day time lag. This diminished





influence suggests that this time period (8-day lag) is the transition point from the wind-resuspension mechanism to the rainfall runoff and river discharge mechanism. Thus, increasing the tropical cyclone size at this 8-day time lag does increase the area affected by tropical cyclone conditions, but these conditions do not necessarily produce the largest nutrient and chlorophyll *a* response at this timescale. Further analysis to identify the exact timing of the change in mechanisms would

require data at an increased temporal resolution which is beyond the scope of this study.

**3.3 Rainfall-River Discharge Mechanism (16 Day Lag)**

More than two weeks after the passage of the tropical cyclone over the coastal lagoon (16-day lag), we again find that tropical cyclone characteristics are significantly related to the variance of chlorophyll *a* concentrations in the Great Barrier Reef: not only through tropical cyclone size, but also through translation speed and rainfall at this time lag. The analysis of

variance shows that the maxima of the characteristics at this time period significantly account for 42-53% of the variance in the mean, 95th percentile, and proportion of area that exceeds 1 mg m$^{-3}$ of chlorophyll *a* (see Table 4 for ANOVA results at 16-day lag). The predictive power is split between size, rainfall, and translation speed, which significantly account for up to 29%, 24%, and 16%, respectively (Table 4). This suggests that at this time lag, storm surge (governed by tropical cyclone size and translation speed) and the catchment inundation (governed by rainfall and tropical cyclone size) cause a greater

amount of change in the surface chlorophyll *a* concentrations than wind speed, which does not significantly account for any of the chlorophyll *a* variance (Table 4).

Over two weeks after tropical cyclone activity (16-day lag), single-factor regressions demonstrate that tropical cyclone wind speeds yield few significant correlations with chlorophyll *a*. Additionally, the positive correlation with minimum values is no longer detected at this longer time lag (Table 5c; Figure 6b,c). This suggests that wind speed and associated benthic re-

suspension no longer excite a response of phytoplankton activity over the Great Barrier Reef 16 days after a cyclone emerges in the area.

We do not find the hypothesized negative correlation with translation speed. There are in fact positive correlations between tropical cyclone translation speed and the maximum values of chlorophyll *a* at a 16-day time lag ($r^2$=0.2268, p=0.029 Table 5c, Figure 7c). While the literature suggests that the contribution of nutrients to the lagoon from storm surge retreat is likely

minor, the findings here could suggest that increasing translation speed increases coastal erosion from storm surge leading to sediment and nutrient loading into the lagoon. The literature suggests that translation speed and storm surge inundation are inversely correlated (Peng et al. 2004; Weisberg and Zheng 2006). However, the suggested inverse relationship may be non-linear (Irish and Resio 2010) and the sensitivity of storm surge to translation speed varies greatly between cases (Peng et al. 2004). Resonance may occur for certain tropical cyclone translation speeds (Rego and Li 2009), whereby increasing

translation speed increases peak surge heights by up to 40% (Rego and Li 2009). Despite flooding a narrower section of the coast, a tropical cyclone with an increased translation speed would move quickly across the shoreline, generating higher surges (Rego and Li 2009). The increase in surge height and associated coastal zone erosion may result in augmented nutrient and sediment leaching from the area affected and thus increased water quality deterioration and phytoplankton





response after the inundation water has retreated back to the lagoon. This hypothesis accounts for the positive correlations between tropical cyclone translation speed and the maximum values of chlorophyll $a$ at a 16-day time lag, and also explains the lack of correlation with the proportion of the area negatively affected (% > 1 mg m$^{-3}$) (Table 5c), since the area inundated would be smaller with increased translation speed. In Figure 7, there appears to be a breakpoint in the data, such that there

are two clusters of data on either side of tropical cyclone translation speed ~12 km h$^{-1}$. We have tested a piece-wise regression considering each individual cluster; however, that approach does not yield a significant relationship between tropical cyclone translation speed and chlorophyll $a$.

There are stronger single-factor positive correlations between tropical cyclone-associated rainfall and chlorophyll $a$ concentrations than with wind speed at 8- and 16-day time lags (Figures 9b,c compared to Figures 6b,c). This relationship is

most pronounced for the extreme chlorophyll $a$ values. While the lag time of river discharge following a tropical cyclone event depends on many factors, the data here suggest that it takes one-to-two weeks for the rainfall recorded along the coastline to move through the catchments, leach nutrients and sediments into the rivers and tributaries, and transport and deposit them in the Great Barrier Reef lagoon, where the phytoplankton response to this influx can be detected by the MODIS satellite. Rainfall runoff and river discharge at these time scales have stronger influences on the chlorophyll $a$

extreme values (and therefore water quality) than wind-driven re-suspension. The correlations found with rainfall were weaker than anticipated; it may be necessary to include catchment-integrated rainfall and stream flow data to fully explore the rainfall runoff mechanism, which is beyond the scope of this study.

At the 16-day lag, tropical cyclone size (radius) still strongly influences chlorophyll $a$ measures such as mean, extreme values, and the proportion of pixels > 1 mg m$^{-3}$ (see Figure 8c for the secondary peak in the correlation with % pixels > 1 mg

m$^{-3}$ at 16-day time lag). As tropical cyclone size increases, the total area from which nutrients and sediments can be leached increases. Storm surge and coastal flood inundation volume increases (Irish et al. 2008), and the total coastal area exposed to heavy rainfall also increases. Subsequently, the positive correlations between these physical effects and nutrient and sediment loading would intensify, ultimately resulting in increased chlorophyll $a$, particularly the extreme values. Given the relationships found between tropical cyclone characteristics and chlorophyll $a$ at this time lag (16 days), we can infer that the

rainfall runoff, river discharge, coastal inundation, and nutrient loading are the main mechanisms affecting the water quality over the Great Barrier Reef, and that the re-suspension mechanism no longer has a role in reducing water quality (Figure 1).

Although some of the correlation values are relatively small, the p-values demonstrate their significance in these processes. Furthermore, a singular tropical cyclone characteristic, size, can account for up to 30-45% of the variability in the surface

water chlorophyll $a$ concentration. In contrast to Brodie et al. (2012a), which demonstrates that the residence time of nutrients and sediments is on the order of a few weeks in the Great Barrier Reef lagoon, we have found no systematic response of the chlorophyll $a$ to tropical cyclone-related weather at timescales longer than 16 days. We find that the wind-driven re-suspension mechanism only contributes to raising minimum values of chlorophyll $a$ in the Great Barrier Reef during and just after the tropical cyclone activity, and that the mean and extreme values which degrade the water quality for





the reef environment appear to be governed primarily by the catchment inundation and river discharge mechanism through tropical cyclone size and, to a lesser extent, translation speed and rainfall at a 16-day lag. However, it is important to note that any external nutrients and sediments brought to the Great Barrier Reef through storm surge and rainfall runoff that are not biologically utilized during the wet season may persist in the lagoon into the dry season (Brodie et al. 2012a). These

latent nutrients may then be re-suspended in the water column through the wind-driven mechanisms during any high-wind event in the dry season, causing an unusual phytoplankton response for the dry season (Brodie and Furnas 1996; Devlin et al. 2001; Udy et al. 2005). In this way, tropical cyclone activity during the wet summer season may still act to reduce water quality in the dry season.

## 4 Conclusions

The results of this study demonstrate that tropical cyclone activity significantly reduces water quality over the Great Barrier Reef at 8- and 16-day time lags. During and just after the passage of the tropical cyclone (no lag and 8-day lag), the wind-driven re-suspension pathway is the dominant mechanism for inducing a phytoplankton response. However, wind speed in isolation only increases minimum levels of chlorophyll $a$ and has no significant effect on the mean or extreme upper values. As such, it may serve to slightly reduce the water quality and render the area more vulnerable to subsequent negative impacts

of runoff. At 16-day time lags, we find a positive influence of tropical cyclone translation speed and coastal rainfall on the mean and extremes of chlorophyll $a$ concentration, but no relationship with wind speed. Hence we suggest that the mechanisms of rainfall runoff, river discharge, and coastal inundation retreat are the dominant factors in reducing water quality in the Great Barrier reef at this time lag. The resulting elevations in mean and extreme measures of chlorophyll $a$ may ultimately lead to negative effects for the coral organisms and ecosystems. These results are consistent with the proposed

mechanism in the literature in which an influx of nutrients and sediments from enhanced erosion and fertilizer loss from the catchment (Brodie et al. 2012b) is drives reduced water quality and elevated phytoplankton activity (Furnas 2003). The mechanism of re-suspension only redistributes the minimal nutrients already in the nutrient-limited ecosystem, which increases minimum chlorophyll $a$ values at an immediate timescale, but is insufficient to cause a more notable reduction in the water quality. The positive correlations between rainfall and chlorophyll $a$ were weak, and hence our results do not

adequately explain the rainfall runoff and river discharge mechanism. In future work, it will be necessary to include catchment-integrated rainfall, stream flow, and wave height data with increased temporal resolution to fully explore the partitioning of the re-suspension, rainfall runoff, river discharge and coastal inundation runoff mechanisms, and the time scales at which these occur.

Our analyses have systematically demonstrated the dominance and significance of tropical cyclone size (radius) in influencing means and extremes of chlorophyll $a$ in addition to the proportion of pixels that exceed 1 mg m$^{-3}$. These effects were observed during the lifecycle of the tropical cyclone, as well as at 8- and 16-day lags. As tropical cyclone size increases, so too do the measures of chlorophyll $a$. Results suggest that the total area affected and area that nutrients can be



extracted from is more important than the duration and intensity of the exposure to tropical cyclone conditions such as wind speed and rainfall.

Our findings support policy recommendations for the reduction of runoff, erosion, fertiliser loss, and nutrient and sediment leaching into the reef lagoon from the Queensland coastal lands, such as the Reef Water Quality Protection Plan (Department of Premiers and Cabinet 2013) and the Paddock to Reef program (Department of Premiers and Cabinet 2013). Our results demonstrate that urban and agricultural areas in catchments that drain into the Great Barrier Reef lagoon need to be particularly vigilant in reducing leaching as urbanization and agricultural intensification continue to affect the reef ecosystem, despite increased awareness of its impacts on the reef ecosystems (see Brodie et al. 2012b for an overview of management responses to nutrient loading of the Great Barrier Reef). These measures will be particularly important in the face of climate change, where global pressures such as ocean warming and acidification are likely to increase, resulting in further coral bleaching, dissolution, and reduced ecosystem resilience to disturbances such as tropical cyclones (Hughes et al. 2003; Hoegh-Guldberg et al. 2007; Pandolfi et al. 2011). Furthermore, while tropical cyclone response to climate change is still an active area of inquiry, it has been suggested that the frequency of intense tropical cyclones (Category 4 and 5) and their associated wind speeds, power dissipation, and damage will increase with rising global temperatures (Bender et al. 2010; Knutson et al. 2010; Done et al. 2012; Holland and Bruyère 2013; Knutson et al. 2013). The results of this study demonstrate that the change to tropical cyclone size with climate change has the potential to influence future water quality of the Great Barrier Reef. Although the size response to climate change is inconclusive thus far in the scientific literature, it has been suggested that tropical cyclone-associated precipitation will increase with climate change (Knutson et al. 2013). Finally, increasing tropical cyclone precipitation would provide a potential increase in nutrient leaching through the rainfall runoff and river discharge mechanism and subsequent increases in phytoplankton response and reductions in water quality. In summary, these relationships suggest that tropical cyclone size and precipitation following landfall are the most critical characteristics to assess in future scenario-building for reef resilience.

**Data Availability**

The Aqua MODIS Level 3 Chlorophyll *a* 4km gridded data set is available at: https://oceancolor.gsfc.nasa.gov/cgi/l3

The tropical cyclone activity in the area can be retrieved from the Australian Bureau of Meteorology Database of past tropical cyclone tracks: http://www.bom.gov.au/cyclone/history/

The tropical cyclone characteristic data is described in Holland et al. 2016 and Done et al. 2015

The coastal weather station data can be retrieved from the Australian Bureau of Meteorolog Weather Station Directory: http://www.bom.gov.au/climate/data/stations/

Analysis was carried out with ENVI, ArcGIS, STATA version 14.1, and R version 3.3.0.





## Author Contributions

C. Parker designed the overall study. C. Parker and S. Spera designed the remote sensing and GIS analysis techniques. K. Spangler aided data organization and design of statistical analysis techniques. C. Parker carried out all analysis and produced all figures. A. Lynch, S. Spera, and K. Spangler contributed to the interpretation of findings. C. Parker prepared the manuscript with contributions from all co-authors.

## Competing Interests

The authors declare that they have no conflict of interest.

## Acknowledgements

The authors would like to thank Daniel P. Moriarty III for copy editing, and James Done and Greg Holland at the National Center for Atmospheric Research (Boulder, CO) for advice and their Cyclone Damage Potential dataset. We also acknowledge the Bureau of Meteorology Australia for providing tropical cyclone data, coastal weather station wind speed and rainfall data, and NASA Ocean Color group for the gridded surface chlorophyll *a* concentration product from the MODIS satellite.

This work was partially supported by a research award from the Institute at Brown for Environment and Society.

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





**Tables**

| Independent Variable | Dependent Variables | Time Range | Statistical Test |
|---|---|---|---|
| Tropical Cyclone Presence / Absence | minimum, mean, 90th percentile, 95th percentile, maximum, and %>1mgm$^{-3}$ | All dates | Students two-sample T-test |
| Tropical Cyclone Size | minimum, mean, 90th percentile, 95th percentile, maximum, and %>1mgm$^{-3}$ | When tropical cyclone > Cat 1 present | Single factor and piece-wise linear regression |
| Tropical Cyclone Maximum Wind | minimum, mean, 90th percentile, 95th percentile, maximum, and %>1mgm$^{-3}$ | When tropical cyclone > Cat 1 present | Single factor regression |
| Tropical Cyclone Translation Speed | minimum, mean, 90th percentile, 95th percentile, maximum, and %>1mgm$^{-3}$ | When tropical cyclone > Cat 1 present | Single factor and piece-wise linear regression |
| Weather Station Rainfall | minimum, mean, 90th percentile, 95th percentile, maximum, and %>1mgm$^{-3}$ | When tropical cyclone > Cat 1 present; When any tropical cyclone present; All dates | Single factor regression |
| Weather Station Wind and Gust Speed | minimum, mean, 90th percentile, 95th percentile, maximum, and %>1mgm$^{-3}$ | When tropical cyclone > Cat 1 present; When any tropical cyclone present; All dates | Single factor single factor regression |
| Tropical Cyclone Wind, Size, Translation Speed, Coastal Rain | minimum, mean, 90th percentile, 95th percentile, maximum, and %>1mgm$^{-3}$ | When tropical cyclone > Cat 1 present | Multivariable ANOVA |

**Table 1: The independent and dependent variables, the time range that was used, and the statistical tests that were used to assess the relationships between the variables.**



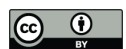

| Presence/Absence | Chlorophyll $a$ Difference (mg m$^{-3}$) (p value) | | | | |
|---|---|---|---|---|---|
| | 0 day lag | 8 day lag | 16 day lag | 24 day lag | 32 day lag |
| Minimum | 0.0015 (0.49) | 0.0005 (0.83) | 0.0002 (0.93) | -0.0033 (0.13) | **-0.0045 (0.04)** |
| Mean | 0.0327 (0.27) | **0.0658 (0.02)** | 0.0087 (0.76) | 0.0237 (0.42) | -0.1344 (0.65) |
| 90$^{th}$ percentile | 0.0118 (0.90) | **0.1998 (0.02)** | 0.0283 (0.75) | 0.0760 (0.40) | -0.0426 (0.64) |
| 95$^{th}$ percentile | 0.1122 (0.40) | **0.3567 (0.01)** | 0.0344 (0.79) | 0.1743 (0.19) | 0.0437 (0.74) |
| %pixels > 1mg m$^{-3}$ | 0.0377 (0.97) | 1.5813 (0.09) | 0.2923 (0.76) | 1.0454 (0.27) | 0.1769 (0.85) |
| Maximum | 2.4533 (0.27) | **7.0814 (0.01)** | **8.0624 (0.003)** | 2.8324 (0.30) | 2.0229 (0.46) |

**Table 2: The T-test results of the difference between chlorophyll a dependent variables given whether a tropical cyclone was present or absent over the reef at varying time lags (calculated by present – absent). The significant results are in bold.**





| | $R^2$ | Proportion of variance (%) | | | |
| --- | --- | --- | --- | --- | --- |
| | | Mean Radius | Mean Translation Speed | Mean Wind | Mean Rain |
| Minimum | 0.099 | 3.532 | 0.049 | 0.002 | 4.740 |
| Mean | *0.568\** | *51.055* | 2.459 | *12.434* | 6.655 |
| 90th percentile | *0.651\** | *60.239* | 1.269 | *14.489* | 5.253 |
| 95th percentile | *0.609\** | *48.329* | 7.353 | 8.718 | 6.785 |
| %pixels > 1mg m$^{-3}$ | *0.548\** | 50.550 | 0.611 | 12.380 | 6.875 |
| Maximum | 0.323 | 3.532 | 23.761 | 0.691 | 10.011 |

**Table 3: The results of the multivariate ANOVA with the averages of tropical cyclone characteristics over the time period as predictor variables and the chlorophyll a measures as the dependent variables at no lag. If the ANOVA overall model is significant (p-value is <0.05), the ANOVA r2 (second column) is given in bold, italic with a star. The proportions that the characteristics account for in the predictive model are given in the subsequent columns. If the relationship is significant (p <0.05) then the percentage is given in bold, italics. The tropical cyclone size significantly accounts for the largest proportion of the predictive power.**





| | | Proportion of variance (%) | | | |
|---|---|---|---|---|---|
| | $R^2$ | Max Radius | Max Translation Speed | Max Wind | Max Rain |
| Minimum | *0.424** | 2.052 | 1.513 | 14.390 | *23.572* |
| Mean | *0.526** | *25.679* | *14.427* | 2.540 | *20.181* |
| 90th percentile | 0.393 | 22.828 | 10.991 | 4.661 | 13.218 |
| 95th percentile | *0.528** | *25.095* | *15.953* | 2.757 | *19.891* |
| %pixels > 1mg m$^{-3}$ | *0.497** | *29.432* | 9.822 | 3.048 | 12.586 |
| Maximum | 0.326 | 1.814 | 26.775 | 2.492 | 9.159 |

**Table 4: The results of the multivariable ANOVA with the maximums of tropical cyclone characteristics over the time period as predictor variables and the chlorophyll a measures as the dependent variables at 16 day lag. If the ANOVA overall model is significant (p-value is <0.05), the ANOVA r2 (second column) is given in bold, italic with a star. The proportions that the characteristics account for in the predictive model are given in the subsequent columns. If the relationship is significant (p <0.05) then the percentage is given in bold, italics. The tropical cyclone size, translation speed and rainfall significantly account for proportions of the predictive power, while wind speed is not significant.**





| (a) | Wind Speed | | Radius | | Translation Speed | |
|---|---|---|---|---|---|---|
| **No Lag** | **Maximum** | **Mean** | **Maximum** | **Mean** | **Maximum** | **Mean** |
| Minimum | 0.022 | 0.026 | 0.013 | 0.037 | 0.001 | -0.011 |
| Mean | 0.025 | 0.026 | *0.253\** | *0.345\** | 0.033 | 0.001 |
| 90th percentile | 0.035 | 0.034 | *0.335\** | *0.433\** | 0.029 | 0.000 |
| 95th percentile | 0.034 | 0.052 | *0.316\** | *0.380\** | 0.094 | 0.026 |
| %pixels > 1mg/m3 | 0.019 | 0.025 | *0.240\** | *0.340\** | 0.008 | -0.002 |
| Maximum | -0.015 | 0.002 | -0.005 | -0.022 | 0.023 | 0.143 |

| (b) | Wind Speed | | Radius | | Translation Speed | |
|---|---|---|---|---|---|---|
| **8 Day Lag** | **Maximum** | **Mean** | **Maximum** | **Mean** | **Maximum** | **Mean** |
| Minimum | 0.174 | **0.262\*** | 0.093 | 0.122 | 0.029 | 0.001 |
| Mean | 0.167 | 0.184 | *0.207\** | 0.185 | 0.067 | 0.041 |
| 90th percentile | 0.151 | 0.169 | *0.271\** | *0.235\** | 0.054 | 0.027 |
| 95th percentile | 0.166 | 0.176 | *0.202\** | 0.151 | 0.077 | 0.072 |
| %pixels > 1mg/m3 | 0.179 | 0.183 | *0.262\** | *0.219\** | 0.043 | 0.024 |
| Maximum | 0.003 | 0.003 | 0.060 | 0.088 | 0.065 | 0.062 |

| (c) | Wind Speed | | Radius | | Translation Speed | |
|---|---|---|---|---|---|---|
| **16 Day Lag** | **Maximum** | **Mean** | **Maximum** | **Mean** | **Maximum** | **Mean** |
| Minimum | 0.052 | 0.062 | 0.003 | 0.000 | -0.047 | -0.081 |
| Mean | 0.126 | 0.181 | *0.369\** | *0.370\** | 0.095 | 0.099 |
| 90th percentile | 0.053 | 0.103 | *0.248\** | *0.242\** | 0.065 | 0.103 |
| 95th percentile | 0.113 | 0.159 | *0.343\** | *0.351\** | 0.109 | 0.108 |
| %pixels > 1mg/m3 | 0.142 | *0.195\** | *0.439\** | *0.404\** | 0.072 | 0.098 |
| Maximum | 0.007 | 0.000 | 0.079 | 0.056 | *0.227\** | *0.190\** |

**Table 5: The R2 values of the single factor regressions with the independent variables of tropical cyclone wind speed, size (radius), and translation speed and the chlorophyll a dependent variables at (a) no time lag, (b) 8-day time lag, (c)16-day time lag. Significant regression values (p < 0.05) are presented in bold and italics, with a star. Size (radius) has a strong positive correlation with chlorophyll a at all three time lags, but it is predominantly strongest at the 16-day time lag, particularly for the mean and % > 1mg/m3.**




**Figures**

Figure 1

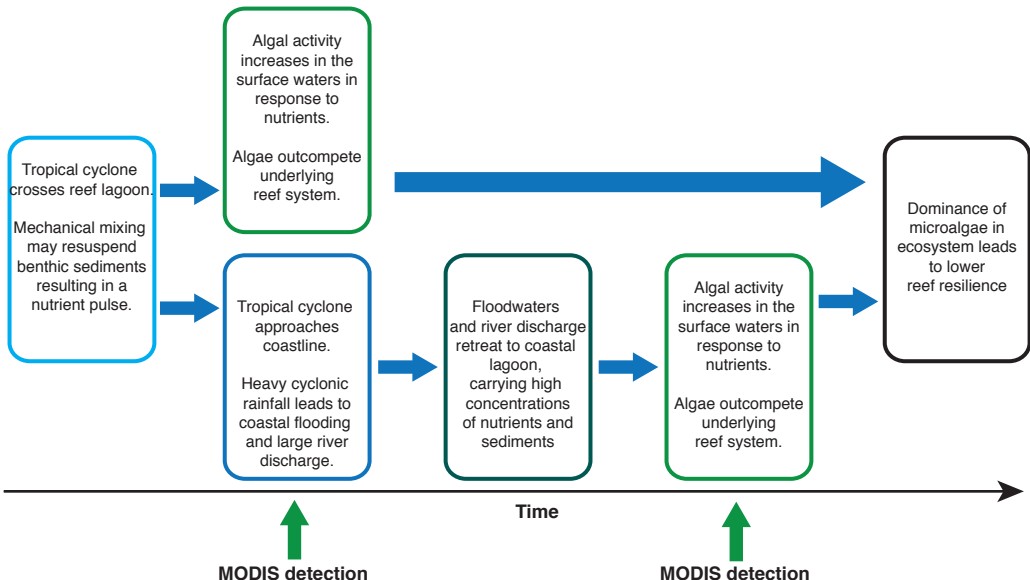

**Figure 1: Postulated mechanisms by which a chlorophyll a response can be detected remotely following tropical cyclone activity.**



Figure 2

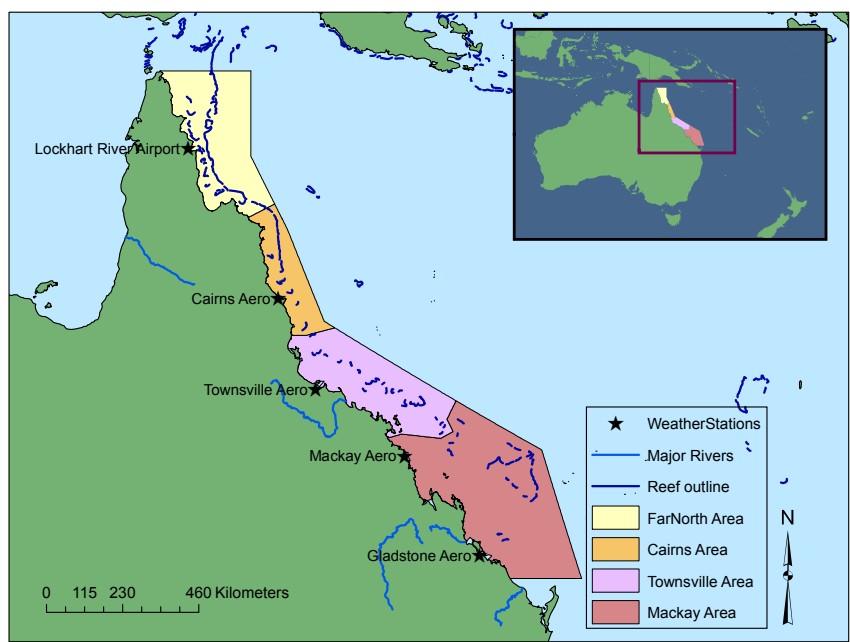

**Figure 2: Great Barrier Reef study area and its designated regions (colored zones). Major reef areas and atolls are outlined in dark blue and major rivers are shown in light blue. The locations of the coastal weather stations are denoted by stars.**





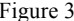

Figure 3

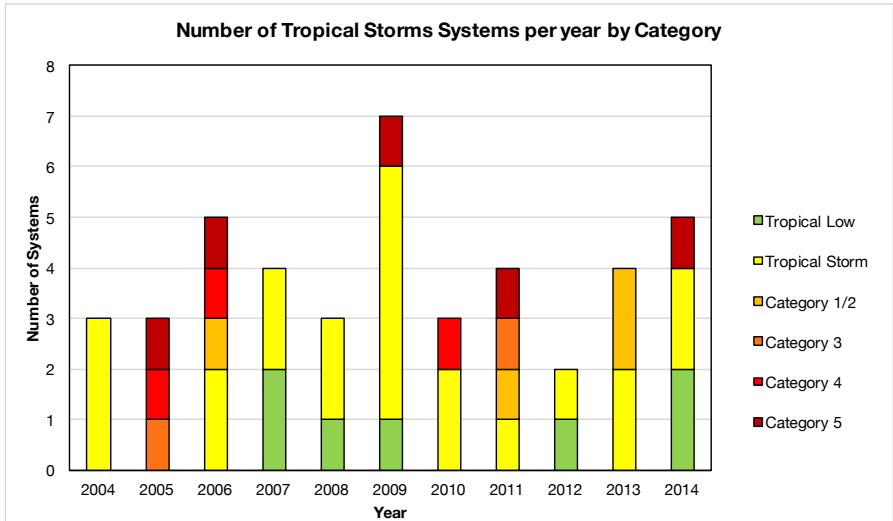

**Figure 3: Tropical storms systems by year separated by Saffir Simpson wind based intensity index. The total number of events in the study period is 43. The storms within a year may overlap in time and thus not produce unique effects on the water quality.**





Figure 4

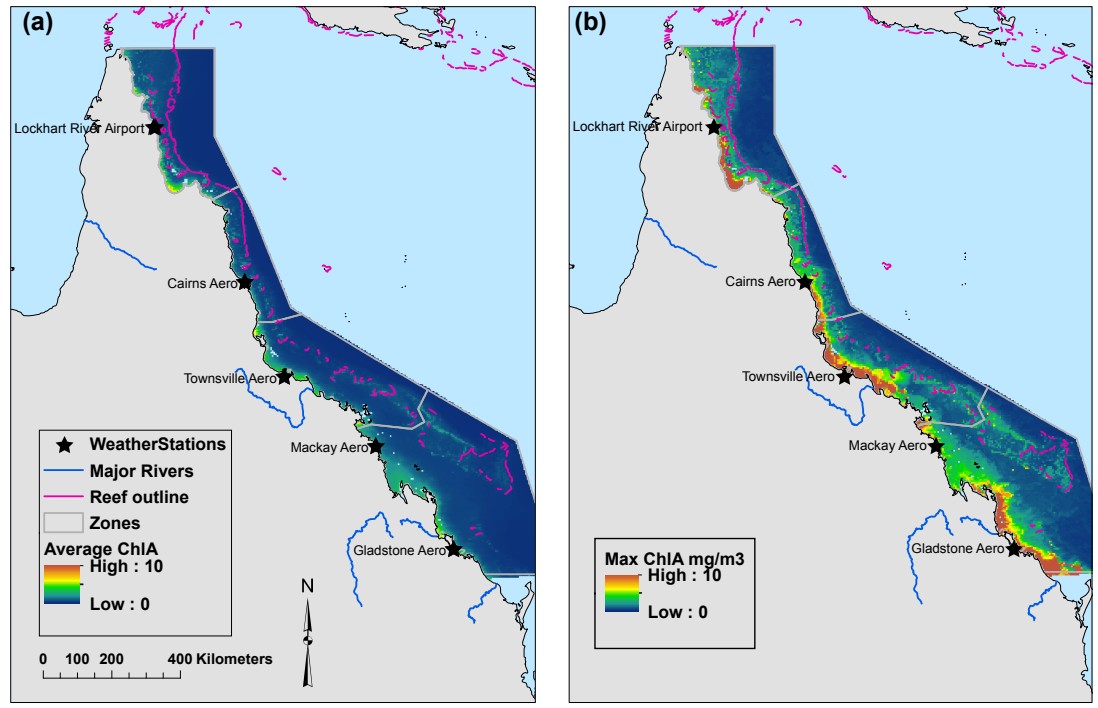

**Figure 4: (a) Average and (b) maximum chlorophyll a concentration, recorded over the 10 year period. Green denotes the 1mg m-3 threshold and yellow and brown denotes exceeding the 1mg m-3 threshold. White pixels denote insufficient clear days to obtain a reliable value.**




Figure 5

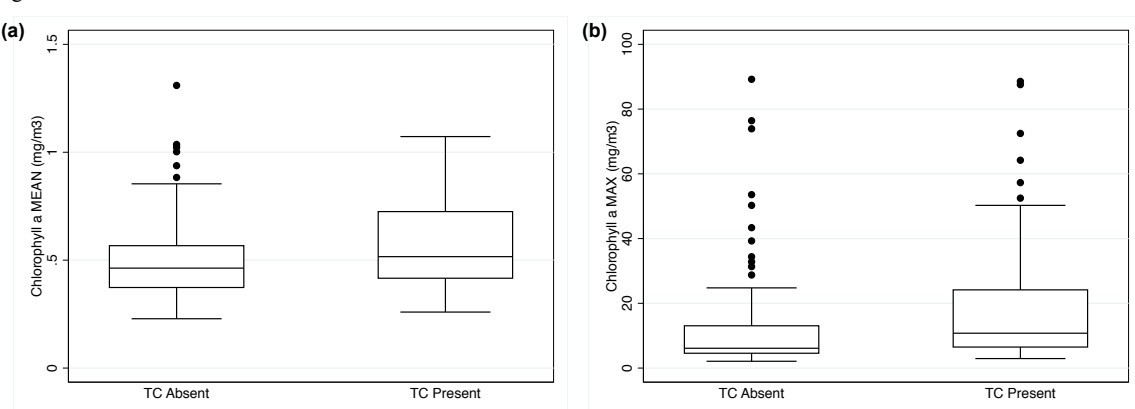

**Figure 5: (a) The spread of the mean chlorophyll a measured over the whole Great Barrier Reef during each time period over the 10-years binned by whether there was a tropical cyclone present or absent in the previous 8-day period. (b) The spread of**
5  **maximum chlorophyll a values recorded over the Great Barrier Reef binned by whether there was a tropical cyclone present or absent in the previous 16-day period.**



Figure 6

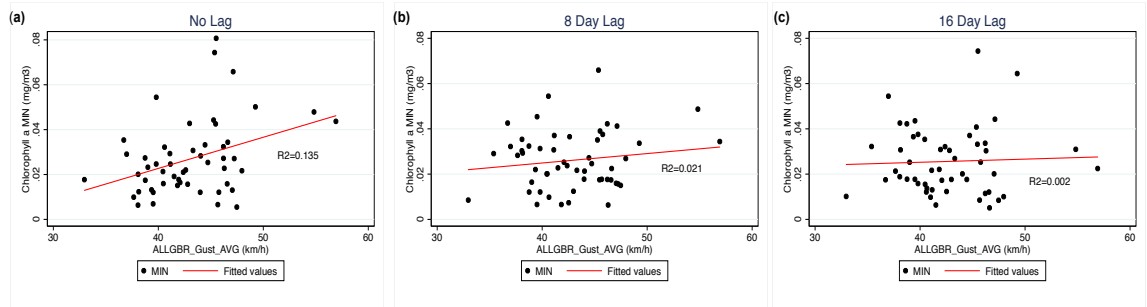

**Figure 6: the minimum chlorophyll a concentration over the Great Barrier Reef against the average coastal gust speed recorded over the time period during tropical activity at a) no time lag b) 8-day time lag c) 16-day time lag to tropical cyclone activity. The line of best fit for the scatter plot is given in red and the R2 value given against the line. The strongest correlation is at no time lag and there is no notable correlation thereafter.**





Figure 7

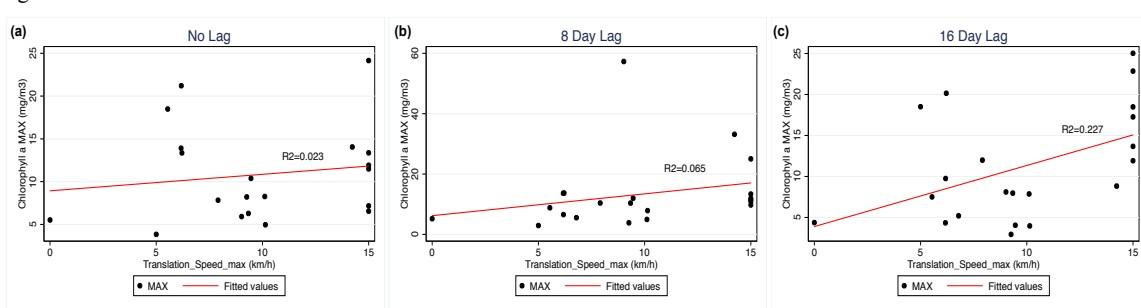

**Figure 7: the maximum chlorophyll a concentration over the Great Barrier Reef against the maximum of the tropical cyclone translation speed over the time period at a) no time lag b) 8-day time lag c) 16-day time lag to tropical cyclone activity. The line of best fit for the scatter plot is given in red and the R2 value given against the line. The R2 values and p-value significance can be found in Table 5. The translation speed cannot explain chlorophyll a variance at the shorter time scales, but over 2 weeks, there is a strong positive correlation.**





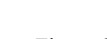

Figure 8

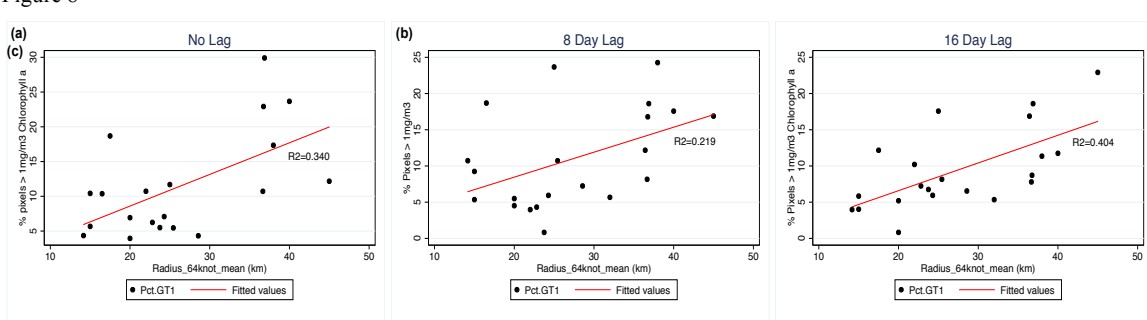

**Figure 8: the % of pixels that exceed the 1mg m-3 threshold over the Great Barrier Reef against the average of the tropical size (radius) over the time period at a) no time lag b) 8-day time lag c) 16-day time lag to tropical cyclone activity. The line of best fit for the scatter plot is given in red and the R2 value given against the line. The R2 values and p-value significance can be found in Table 5. The strongest correlation is at the 16-day time lag. But the positive linear relationship is sustained from no time lag to the 16-day time lag and for many of the important measures of chlorophyll a such as the mean, extremes, or % of pixels > 1mg m-3.**



Figure 9

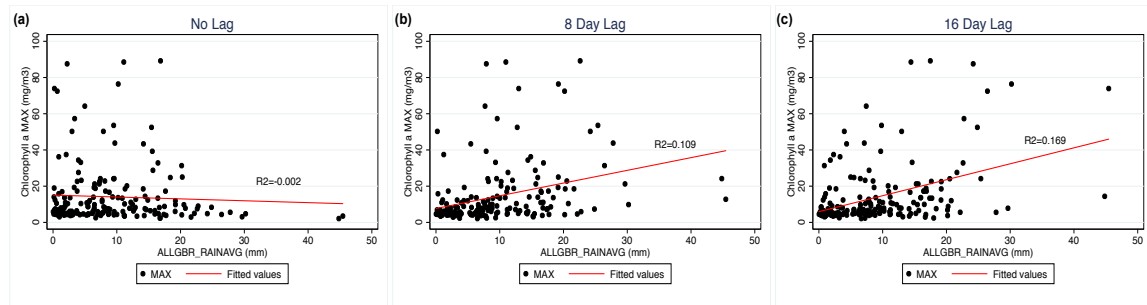

**Figure 9: the maximum chlorophyll a concentration over the Great Barrier Reef against the average rainfall measured along the coastline for the whole 10 year study period at a) no time lag b) 8-day time lag c) 16-day time lag to rainfall activity. The line of**
5   **best fit for the scatter plot is given in red and the R2 value given against the line. The positive correlation is found at the 8-day time lag increases to the 16-day time lag.**