# Peer review of "The Relationship between Tropical Cyclone Activity, Nutrient Loading, and Algal Blooms over the Great Barrier Reef"

_Biogeosciences, 2017_

## Referee Comment (RC1) · Anonymous Referee #1 · 14 Mar 2017

Review on bg-2017-23 This study evaluates the effects of tropical cyclones, either directly by sediment re-suspension or by indirectly by the rainfall and river discharge, on phytoplankton in the surface waters of the Great Barrier Reefs. It is an important issue in oceanography and coral reef protection and well within the purview of BG. However, there are many flaws and ambiguities that the conclusions can be made, I do not recommend a publication on the current version.

General comments: 1. The accuracy or uncertainty of the remotely sensed chlorophyll a concentration in this study area should be quantitatively evaluated. According to Udy et al. (2005), the correlation between the remotely sensed chlorophyll a concentrations and the field in situ measurements in this area seemed insignificant. If the

remotely sensed chlorophyll a concentration can not represent the ground-true values, any statistics based on the former should not be accepted.

2. Since the affected area by a tropical cyclone is limited, it is better to evaluate the effects of the tropical cyclone event-by-event, but not taking the GBR as a whole. For example, considering two tropical cyclones with the same intensity, size, moving speed, etc., one that passes over a catchment area and thus induces a stronger river plume should have a stronger effect than the other one that propagated far away from the major rivers. Moreover, if the tropical cyclone-induced rainfall and river discharge are so important, there should have a gradient with the decreasing effect of the tropical cyclone with the increasing distance leaving from the land. Many studies have shown that the inshore chlorophyll a concentrations are generally higher in the wet season than in the dry season, but the seasonal difference may become insignificant in the offshore areas.

Specific comments: 1. P2, L8-19: At the first few days, in addition to the stimulated phytoplankton growth by the input nutrients, the elevated chlorophyll a concentrations in the surface waters may be, at least partly, from the vertical mixing which brings the high-level chlorophyll waters in the lower layer to the surface waters.

2. P2, L17-19: I don't think the sediment loading has to be collinear with chlorophyll a concentration. Although the nutrient loading may be associated with the sediment loading and this may stimulate the phytoplankton growth, the reduced light intensity associated with the sediment loading may limit the phytoplankton growth.

3. P3, L28-33: In those exampled coastal regions (e.g. the Chesapeake Bay), the accuracy and uncertainty of the remotely sensed chlorophyll a concentrations have been evaluated extensively.

---

## Referee Comment (RC2) · Anonymous Referee #2 · 17 Mar 2017

An interesting and useful idea to try to assess the extent to which cyclones may be affecting water quality across the GBR. However, I agree with Reviewer #1 that an essential precursor to your study is an accuracy assessment of the MODIS product for GBR waters. I further agree with Reviewer #1 that it makes much more sense to consider each cyclone separately and at the appropriate spatial scale, as I'll comment on below.

Specific comments:

As far as I can tell, you measured storm activity as present or absent anywhere in the GBR based on whether a cyclone track crossed the Lagoon. Or did you do this for every 4km pixel across the GBR? However a given eye position along a cyclone track

can be located outside the GBR while its winds and waves penetrate within, depending on its position, intensity, translation speed and circulation size. Thus, you could entirely miss a relevant cyclone. As luck would have it, no such cyclones happened to occur during your study period, but have done so regularly prior to 2004 and could do in future.

I am surprised that the MODIS 8 day product only provides data for an arbitrarily set 8 day period, rather than data for every day based on input from the 8 preceding days or based on the 8 following days.

The necessity of analysing the data in set 8 day blocks introduces uncertainty into the time lag analysis which make it difficult to be certain what the time lags actually represent. For example, if a cyclone affected the GBR within a single day (often the case) and this day happened to be day 1 of the 8 day block, the effective time lag being measured for that storm in the 8 day lag scenario could be up to 15 days even though the analysis suggests it was only 8. Similarly, if a cyclone affects the GBR on day 8 of the 8 day block and the corresponding reduction in water quality is evident in the 8 day lag block, the actual time difference from cyclone to lower water quality could still be a little as a day if the response occurred on day 1 of the lagged 8 day block. When analysing the data statistically, you don't have the ability to account for this which makes it hard to be confident with the associated stats. This seems to confound the ability to be sure that a given Chl A response was caused by wind driven suspension versus a flood plume.

Wind-driven resuspension would be caused by the locally generated wind-sea not the wind itself. Since the magnitude of locally generated seas that form in a given location depends on both wind speed and the duration of those winds, I would have been surprised if wind speed alone was important especially given you did not map the spatial extent of the wind speeds. The max wind speed of a cyclone is typically concentrated in a very narrow band at the eyewall and the bulk of the wind speeds are much lower because wind speed drops quickly with distance from the eye (see Holland, Greg J.

"An analytic model of the wind and pressure profiles in hurricanes." Monthly weather review 108.8 (1980): 1212-1218). In large cyclones, this drop-off occurs more gradually, spreading higher winds over a larger area. It thus makes sense that size would be more important than max wind speed. However, I would expect that it would be the interplay between size, intensity and translation speed that would make the most difference – a large, strong cyclone that moves slowly has the greatest potential to generate the largest local seas for a given maximum wind speed. Also it is very important where the cyclone tracked because the distance of water over which wind of a given speed blows (fetch) is also key to what waves develop. This is why it would make sense to map the spatial extent of the wind fields (for example, see Puotinen, Marji, et al. "A robust operational model for predicting where tropical cyclone waves damage coral reefs." Scientific reports 6 (2016)) rather than just using maximum values for an entire cyclone.

TC rainfall is not evenly distributed around the track and the size of the area where it occurs varies with intensity (for example, see Manuel Lonfat, Frank D. Marks Jr., and Shuyi S. Chen. (2004) Precipitation Distribution in Tropical Cyclones Using the Tropical Rainfall Measuring Mission (TRMM) Microwave Imager: A Global Perspective. Monthly Weather Review 132:7, 1645-1660.) For example, the area over which heavy rainfall occurs contracts significantly as a cyclone intensifies. This may help explain why size was a more important predictor than intensity – even though max rainfall rate is positively correlated with intensity, such rainfall would be spread over a larger area for a bigger storm.

As far as I can tell, you attempted to relate cyclone characteristics (size, intensity, translation speed) to Chl A concentrations without considering the spatial extent of heavy rainfall for a given cyclone at a given time with respect to the area of catchment. First, the cyclone characteristics typically vary considerably along the track so if you used averages for an entire cyclone, it could be very misleading. Further, the spatial extent of area exposed to TC activity will vary along the track, often quite considerably. You

could map the spatial zone where cyclone rainfall would be expected to occur using the distance to 64 knot winds (or better the distance to gales) every hour (or at least every 6 hours). From these data, you could calculate the % area of each catchment that coincides with a given cyclone each time step and then determine the total number of hours that rain likely fell across the catchment. Then you could determine whether raised ChlA levels seen in the GBR Lagoon at the time of a given cyclone seem plausible given which catchment likely experienced rain and by how much. Not doing this means you don't know how likely it was that a flood plume entered the GBR Lagoon from a given cyclone and where it was likely to occur, and it means you don't have a basis other than the time-lag analysis upon which to ascertain whether the Chl A response was due to wind driven resuspension versus flood plumes.

You should note that to really be confident that a Chl A response you observe is due to a cyclone flood plume would require modelling the location of such flood plumes over time (and cite the relevant studies) even though that was beyond the scope of your study.

---

## Author Comment (AC1) · 2 May 2017

Response to Reviewers

Title: The Relationship between Tropical Cyclone Activity, Nutrient Loading, and Algal Blooms over the Great Barrier Reef Author(s): Chelsea L. Parker, Amanda H. Lynch, Stephanie A. Spera, Keith R. Spangler MS No.: bg-2017-23

VALIDATION

We appreciate the concerns of the reviewers regarding the accuracy and reliability of the MODIS chlorophyll a data in the coastal region of the Great Barrier Reef (GBR). It is evident that validation of this dataset needs to be carried out.

**BGD**

In response to these concerns, we have begun validation of the re-motely sensed data product using chlorophyll a in situ measurements from NOAA cruise data measurement (the World Ocean Database, available at https://www.nodc.noaa.gov/OC5/SELECT/dbsearch/dbsearch.html). Importantly these NOAA cruises sample the GBR marine park and measurements capture Case 2 waters and allow for the validation of the MODIS product in these complex waters. Given the rarity of cruises, there were only 4 passages that crossed the GBR and coincided with available MODIS data in our 10-year study period.

Our initial validation aggregates all the available NOAA WOD data from our time period partitioned by times of TC presence (Figure 1) or absence (Figure 2). Both relation-ships demonstrate a significant positive correlation ($p < 0.05$) between the in situ and the remotely sensed product (Figure 1, 2). The MODIS data reasonably captures the trend in in situ measurements. Figure 1 begins to suggest that the MODIS product may be systematically underestimating surface chlorophyll a concentrations during times of TC presence. This new analysis suggests that the discussed signals and relationships between chlorophyll a and tropical cyclone (TC) activity in the manuscript are still valid, and may even be underestimated.

We could include detailed analysis of this validation effort in the manuscript to address concerns about the MODIS data accuracy in this region. We could also expand the validation to time periods outside of 2004-2014 in order to increase the number of cruises aggregated and systematically demonstrate validation of this MODIS dataset in the GBR region (particularly during times of TC present). This would provide a very important step in the validation of the MODIS product in these waters which is clearly lacking in the literature.

AGGREGATION

We would like to clarify that this study seeks to find a significant, generalizable, sys-tematic relationship between tropical cyclone activity and chlorophyll a over the GBR

in order to generate hypotheses of the mechanisms through which cyclone may affect reef water quality. We explore whether a signal can be found in the data as a precursor to more detailed studies. This study uses broad spatial and temporal scales, and finding a signal at this scale informs us of the utility of progressing to finer scale, detailed studies. Furthermore, this work stands in concert with, and does not aim to recreate the existing smaller scale, case study projects and observations.

In order to explore and identify a signal in the data, it was necessary to use a large number of cyclone cases (for a large sample size) and aggregate the data over 10 years of cyclone activity and over the whole expanse of the GBR. If necessary, we can attempt to make this motivation clearer in the manuscript.

DIRECT RESPONSES

REFEREE 1

General comment 1: we are beginning to quantitatively evaluate the MODIS chlorophyll a concentrations as discussed in the validation section above.

General comment 2: we appreciate these comments and concerns over spatial and temporal scales. As discussed in the aggregation section above, the utility and novelty of this work lies in the exploration of a signal in a large, aggregated dataset. We also agree that there would be a gradient of effect with distance from the land. However, given that we use a range of descriptive statistics including the mean, upper percentiles, and maximum values, the affected zone would be captured in these measures regardless of the location and size of the cyclones. If necessary, we could include a breakdown of the analysis to examine inshore and offshore reef environments to further constrain the mechanisms.

Specific comment 1: we agree that vertical mixing may play a role in elevating chlorophyll a concentrations. This vertical mixing could also be attributed to TC-winds and would still fit within our mechanism framework and we could include a discussion on

this point. However, work and previous comments from Jon Brodie suggest that vertical mixing would be negligible over the GBR area given the depth of the water column in this region.

Specific comment 2: this is an interesting point regarding the co-linearity of sediment loading and chlorophyll a concentrations and we would be interested in looking into it further. However, we suggest that at the ocean surface (where the MODIS measurements are made), the phytoplankton would not be light limited by the suspended sediments further down in the water column. While the relationship may not be exactly 1:1, we maintain that sediment loading is associated with nutrient loading which can stimulate phytoplankton growth – as the reviewer also discusses.

Specific comment 3: we agree that evaluation of MODIS is necessary in this region, therefore we have begun this undertaking as discussed in the validation section above.

REFEREE 2

General comments: we thank the reviewer for their overarching comments. The validation work we are undertaking should address the concerns with MODIS accuracy over the GBR, as discussed above in validation. We would also like to reiterate that the main aim of this work is to explore systematic, generalizable relationship through aggregated data as is discussed in detail in the aggregation section. This is why we have not undertaken analyzing the TC events separately as has been done before in previous papers.

Specific comments:

We did assess the TC activity as whether the cyclone track crossed within the GBR Marine Park. As the reviewer says, during our time period of study we have not missed a relevant cyclone. This is an important point and for other time periods, additional TC systems may have to be assessed.

To the best of our understanding, the MODIS 8-day product does indeed provide the

best available pixel from an 8-day window which is set as a fixed 8-day date by the processors of the data.

We agree that using 8-day time blocks hinders the temporal resolution and perhaps some of the confidence in the exact timings of the mechanisms. Analysis at higher temporal resolution would be hampered by the availability of daily MODIS data. However, it is important to note that TCs do not systematically occur early or late in the 8-day time period, they are randomly distributed. We sought to overcome the additional complexity of where the TC occurs in the time period by using a large number of storm systems over a 10-year period, and aggregating them to find a signal. We find significant relationships with wind speed at no time lag and then no significant relationships at any greater time lags (Section 3.2); and for rainfall and translation speeds we find the reverse, significant correlations only at time lags of 8-days and greater (Section 3.3). This systematic response with aggregated data highlights the validity of the hypothesis of the two mechanisms operating over different time scales and the relative importance of the two that can now be investigated further. Furthermore, we have assessed that the systems do often affect the GBR marine park for typically more than one day and therefore their effect stretches further into the 8-day period that they occur in.

We agree with the discussion that wind-driven resuspension is controlled by wind speed, duration of winds, and area of TC winds. We do in fact include analysis of translation speed to account for duration and radius of 64 knot winds to account for size (Section 2.2). In Section 2.4 we hypothesize that if the wind mixing mechanism and the duration the reef is exposed to TC winds were dominant, we would expect to see a negative correlation of chlorophyll a with translation speed. However, our results show no such negative correlation suggesting (Section 3.2, 3.3). We agree that TC size is affecting the wind driving experienced and we do discuss that the positive correlations between cyclone size and chlorophyll a at no time lag would be caused by a greater area of the reef being affected by TC-conditions–in this case wind mixing at

this time lag (Section 3.2).

Examining the spatial distributions of wind and precipitation fields for each TC would be a very interesting idea. However, we would like to reiterate from the aggregation discussion above that the aim of this study is to determine whether there is a systematic response to TCs and their overall characteristics, regardless of the finer details of the wind and precipitation fields. Using only the time-lag analysis to ascertain whether the chlorophyll a response was due to the wind mixing or runoff and river discharge mechanisms is an efficient and effective method to assess a generalizable relationship between TC activity and water quality over the GBR.

We agree that adding a discussion about the observation of cyclone flood plumes through modelling techniques would be an interesting addition to the paper.

[Figure]

**Fig. 1.** The relationship between the in situ chlorophyll a concentrations measured from NOAA WOD and the MODIS chlorophyll a concentration product during times of TC presence in the area.

[Figure]

**Fig. 2.** The relationship between the in situ chlorophyll a concentrations measured from NOAA WOD and the MODIS chlorophyll a concentration product during times of TC absence in the area.